# Prenatal SARS-CoV-2 Infection Alters Human Milk-Derived Extracellular Vesicles

**DOI:** 10.3390/cells14040284

**Published:** 2025-02-15

**Authors:** Somchai Chutipongtanate, Supasek Kongsomros, Hatice Cetinkaya, Xiang Zhang, Damaris Kuhnell, Desirée Benefield, Wendy D. Haffey, Michael A. Wyder, Gaurav Kwatra, Shannon C. Conrey, Allison R. Burrell, Scott M. Langevin, Leyla Esfandiari, David S. Newburg, Kenneth D. Greis, Mary A. Staat, Ardythe L. Morrow

**Affiliations:** 1Department of Environmental and Public Health Sciences, University of Cincinnati College of Medicine, Cincinnati, OH 45267, USA; kongsosk@ucmail.uc.edu (S.K.); cetinkhe@ucmail.uc.edu (H.C.); zhanx5@ucmail.uc.edu (X.Z.); kuhnelds@ucmail.uc.edu (D.K.); conreysc@ucmail.uc.edu (S.C.C.); newburdd@ucmail.uc.edu (D.S.N.); 2Department of Biomedical Engineering, University of Cincinnati, Cincinnati, OH 45267, USA; esfandla@ucmail.uc.edu; 3Center for Advanced Structural Biology, Department of Molecular & Cellular Biosciences, University of Cincinnati College of Medicine, Cincinnati, OH 45267, USA; benefide@ucmail.uc.edu; 4Department of Cancer Biology, University of Cincinnati College of Medicine, Cincinnati, OH 45267, USA; dominiwd@ucmail.uc.edu (W.D.H.); wyderma@ucmail.uc.edu (M.A.W.); greiskd@ucmail.uc.edu (K.D.G.); 5Department of Infectious Disease, Cincinnati Children’s Hospital Medical Center, Cincinnati, OH 45267, USA; gaurav.kwatra@cchmc.org (G.K.); allison.burrell@cchmc.org (A.R.B.); mary.staat@cchmc.org (M.A.S.); 6Department of Clinical Microbiology, Christian Medical College, Vellore 632004, India; 7Larner College of Medicine, University of Vermont, Burlington, VT 05405, USA; scott.langevin@uvm.edu

**Keywords:** COVID-19, exosomes, human milk, microRNAs, proteomics

## Abstract

Human milk-derived extracellular vesicles (HMEVs) are key components in breast milk, promoting infant health and development. Maternal conditions could affect HMEV cargo; however, the impact of SARS-CoV-2 infection on HMEVs remains unknown. This study investigated the influence of SARS-CoV-2 infection during pregnancy on postpartum HMEV molecules. The median duration from SARS-CoV-2 test positivity to milk collection was 3 months. After defatting and casein micelle disaggregation, HMEVs were isolated from milk samples of nine mothers with prenatal SARS-CoV-2 and six controls by sequential centrifugation, ultrafiltration, and qEV-size exclusion chromatography. The presence of HMEV was confirmed via transmission electron microscopy. Nanoparticle tracking analysis demonstrated particle diameters of <200 nm and yields of >1 × 10^11^ particles per mL of milk. Western immunoblots detected ALIX, CD9, and HSP70, supporting the presence of HMEVs in the isolates. Cargo from thousands of HMEVs were analyzed using a multi-omics approach, including proteomics and microRNA sequencing, and predicted that mothers with prenatal SARS-CoV-2 infection produced HMEVs with enhanced functionalities involving metabolic reprogramming, mucosal tissue development, and immunomodulation. Our findings suggest that SARS-CoV-2 infection during pregnancy boosts mucosal site-specific functions of HMEVs, potentially protecting infants against viral infections. Further prospective studies should be pursued to reevaluate the short- and long-term benefits of breastfeeding in the post-COVID era.

## 1. Introduction

Human milk, a complex and dynamic biofluid, provides nutrients that support infant growth and bioactive components that protect infants against various diseases [1,2], including respiratory infections in early and late childhood [3]. Clinical trials and epidemiologic studies confirm the beneficial effects of feeding human milk over infant formula in preventing early- and long-term diseases [2,3,4]. Cumulative evidence of the mechanisms shows breastfeeding delivers positive health outcomes to children through human milk bioactive components, including oligosaccharides, immunoglobulins, growth factors, cytokines, adipokines, hormones, lipids, peptides, cells, and extracellular vesicles (EVs) [5].

Human milk-derived EVs (HMEVs) are lipid bilayer-enclosed nanoscale vesicles that are mainly secreted by mammary epithelial cells and play roles in physiological functions and pathological processes [5,6,7]. HMEVs carry selective molecular cargo from mammary glands with the potential to modulate gene expression and cell signaling in infant tissues [5,6,7]. Current evidence suggests that HMEVs represent a vital mechanism of mother-to-child intergenerational communication [5,6,7]. The infant intestinal mucosa appears to be a primary target of HMEVs. Appreciable amounts of HMEVs are absorbed from the gastrointestinal tract into the blood circulation and, from there, can reach various organs to modulate functions in recipient cells, as suggested through in vitro and in vivo studies [8,9,10,11]. Research has shown that maternal pathological conditions during pregnancy affect the molecular cargo of HMEVs, with potential functional changes in breastfed infants [5,12,13,14], while the exact mechanisms remain to be elucidated.

The coronavirus disease 2019 (COVID-19) pandemic, caused by severe acute respiratory syndrome coronavirus-2 (SARS-CoV-2) infection, presented a significant challenge to public health globally. While COVID-19 affected breastfeeding practices around the world due to concerns about possible mother-to-infant transmission [15], there is no evidence of direct viral transmission through breastfeeding [16,17]. Maternal exposure to SARS-CoV-2 has been shown to influence human milk components, including antibodies, inflammatory mediators, cytokines, and immune cells [18,19,20,21,22,23,24,25,26,27]. Nevertheless, our knowledge about the impact of maternal SARS-CoV-2 infection on HMEVs remains limited.

The NIH-funded IMPRINT birth cohort collected data on maternal COVID-19 history and maternal blood in the third trimester and implemented a standardized protocol for the collection of breast milk at 2 weeks postpartum and several times thereafter. During the COVID-19 pandemic, a substantial proportion of participating mothers experienced SARS-CoV-2 infection during pregnancy, thereby providing an opportunity to examine the potential effects of prenatal SARS-CoV-2 infections on the constituents of postpartum human milk, inclusive of HMEVs.

Using samples and data from the IMPRINT cohort, the aim of this study was to ascertain the influence of prenatal SARS-CoV-2 infection on HMEV molecular cargo. Following the optimization of our previously established method [28] for the successful isolation of EVs from the complex matrix of human milk, we performed a multi-omic analysis. This incorporated mass spectrometric-based proteomics and miRNA sequencing on HMEVs isolated from nine mothers with prenatal SARS-CoV-2 infection, compared to six healthy control subjects. Alterations in HMEV cargo were detected, and functional predictions were carried out based on those changes to elucidate potential impacts on child health outcomes. Our findings suggest that SARS-CoV-2 infection during pregnancy may boost the mucosal site-specific functions of HMEVs in breastfed infants.

## 2. Materials and Methods

### 2.1. HMEV Isolation

Human milk samples (1 mL/sample) were retrieved from the IMPRINT birth cohort (protocol ID 2019-0629). Milk samples were subjected to stepwise centrifugation (500× *g* for 15 min, 3000× *g* for 15 min) to remove cells and milk fat. Sodium citrate was added to the final concentration of 1% (*w*/*v*) and incubated at 4 °C for 30 min to disrupt casein micelles, and the sample was centrifuged at 14,000× *g* for 30 min to remove larger microvesicles. The supernatant was then concentrated using a 100 kDa cutoff centrifugal filter (Thermo Scientific, Waltham, MA, USA) to achieve 0.5 mL volume. The concentrated supernatant was loaded onto the Izon qEV size-exclusion column (qEVoriginal/35 nm; Izon Science, Oxford OX5 1PF, UK) to isolate the EV particles and remove soluble protein contaminants. The particle eluates were pooled and concentrated by 100 kDa cutoff centrifugal filtration (Thermo Scientific), resulting in a final volume of 1 mL. The Pierce660 assay (Thermo Scientific) was used to estimate EV protein amounts. All isolation procedures were performed at 4 °C or on ice, as appropriate. The isolated EVs were aliquoted (100 μL per aliquot) and kept at −80 °C until use.

### 2.2. Nanoparticle Tracking Analysis (NTA)

A NanoSight NS300 (Malvern Instruments Ltd., Malvern, Worcestershire, UK) was used for particle size distribution and concentration analysis. The sample was diluted 1:1000 in PBS to a final volume of 1 mL. The diluted sample was injected using a 1 mL syringe. Five 1 min videos were captured for each sample by an sCMOS camera with cell temperature maintained at 25 °C. Videos were analyzed by NanoSight software NTA 3.4 Build 3.4.003 with a detection threshold = 5 and blur size and max jump distance set to auto. Ideal concentrations were measured at 20–100 particles/frame.

### 2.3. Transmission Electron Microscope (TEM)

Three microliters of the EV sample were adsorbed onto individual pre-cleaned 300 mesh Formvar/Carbon TEM grids (FCF300-CU_UB; Electron Microscopy Sciences, Hatfield, PA, USA) and negatively stained using freshly prepared 2% uranyl acetate (UA) aqueous solution (#22400; Electron Microscopy Sciences). Briefly, samples were adsorbed for 1 min before blotting away excess sample. The grids were subsequently washed twice in de-ionized water before a final one-minute incubation in a 2% stain step of one minute. Excess stain was blotted away, and the grids were allowed to dry before storing. Imaging was performed using a Talos L120C transmission electron microscope (Thermo Scientific) using an acceleration voltage of 120 kV. Images were acquired with a Ceta-16M CMOS camera (Thermo Scientific) using a one-second exposure time and a total electron dose of ~40 e-/Å^2^ per image.

### 2.4. Western Blot Analysis

The EV sample (10 μg protein) was mixed with laemmli buffer, heated at 95 °C for 10 min, resolved in a 4–12% Invitrogen B-T gel using MOPS buffer, and transferred onto a polyvinylidene difluoride (PVDF) membrane using a Trans-Blot transfer system (Bio-Rad Laboratories, Inc., Hercules, CA, USA). The membrane was blocked with 5% skim milk and then probed with primary antibodies as follows: anti-Hsp70 (1:1000) (#sc-373867; Santa Cruz Biotechnology, Dallas, TX, USA), anti-CD9 (1:1000) (#sc-13118; Santa Cruz Biotechnology), and anti-Alix (1:1000) (#2171; Cell Signaling Technology, Inc., Danvers, MA, USA) antibodies at 4 °C overnight. After washing, the membranes were incubated with anti-mouse IgG-HRP (1:3000) (#NA931, MilliporeSigma, Burlington, MA, USA) at room temperature for 1 h. The immunoblot was developed by SuperSignal West Pico PLUS Chemiluminescent (Thermo Fisher) and imaged on a ChemiDoc Touch Imaging system (BioRad Laboratories, Inc.).

### 2.5. NanoLC-MS/MS

Two micrograms of EV proteins was dried in a speedvac and resuspended in 15 μL of 50 mM ammonium bicarbonate. Samples were then reduced by dithiothreitol (DTT; a final concentration of 10 mM) with heat at 95 °C for 5 min, alkalinized by iodoacetamide (IAA; a final concentration of 10 mM) for 20 min at room temperature in the dark, and digested by trypsin (1:50 *w*/*w*; MS-grade, Thermo Scientific) at 37 °C overnight. Samples were desalted by C18 solid phase extraction, dried by a speedvac, and resuspended in 0.1% formic acid in H_2_O. Data were collected on an Orbitrap Eclipse mass spectrometer coupled to a Dionex Ultimate 3000 RSLCnano (Thermo Scientific). Peptide was injected onto a 5 mm nanoViper μ-Precolumn (internal diameter [i.d.] 300 mm, C18 PepMap100, 5.0 μm, 100 Å; Thermo Scientific) at 5 μL/min in 0.1% formic acid in H_2_O for 5 min. For chromatographic separation, the trap column was switched to align with EASY-Spray column PepMap RSLC C18 with a 150 mm column (i.d. 75 μm, C18, 3.0 μm, 100 Å). Peptides were eluted using a variable mobile phase gradient from 98% phase A (0.1% formic acid in H_2_O) to 32% phase B (0.1% formic acid in acetonitrile [ACN]) for 60 min at 300 nl/min. MS1 was collected in an Orbitrap (120,000 resolution; maximum injection 50 ms; automatic gain control [AGC] 4 × 10^5^). Charge states between 2 and 6 were required for MS^2^ analysis with a 20 s dynamic exclusion window and cycle time of 2.5 s. MS^2^ scans were performed in an ion trap with higher-energy collisional dissociation (HCD) fragmentation (isolation window 0.8 Da; normalized collision energy (NCE) 30%; maximum injection 40 ms; AGC 5 × 10^4^) and recorded using Xcalibur 4.3 (Thermo Scientific). The raw files were searched using Proteome Discoverer v.2.4 (Thermo Scientific) against the human protein database and the Sequest HT search algorithm with LFQ parameter to identify and quantify peptides at FDR < 1% before protein inference (reported at >99% confidence). The results were exported in Excel format for further analysis. Raw expression data were log2 transformed and normalized by VSN (variance stabilization normalization), and missing values were imputed by MinDET (deterministic minimum algorithm) before differential expression analysis and functional prediction.

### 2.6. MicroRNA-Sequencing

NEBNext Small RNA Sample Library Preparation kit (NEB, Ipswich, MA, USA) was used to construct the library with a modification for precise library size selection with high sensitivity [29]. Briefly, after 15 cycles of final PCR, the libraries with unique indices were first equal to 10 µL, pooled, cleaned up, and mixed with a custom-designed DNA ladder containing 135 and 146 bp DNA. This size range corresponds to an miRNA library with a 16–27 nt insert that covers all miRNA. After high-resolution agarose gel electrophoresis, the library pool, which ranged from 135 to 146 bp, including the DNA marker, were gel-purified and quantified by NEBNext Library Quant kit (NEB) using QuantStudio 5 Real-Time PCR System (Thermo Scientific). The first sequencing was performed on a NextSeq 2000 sequencer (Illumina, San Diego, CA, USA) to generate a few million reads to quantify the relative concentration of each library. The volume of each library was then adjusted to generate >3 M reads per sample in the second sequencing for final data analysis. The raw fastq files were processed by Illumina Sequence Hub-miRNA analysis: trim adapter using cutadapt, map trimmed reads on miRNA precursors using SHiMPS aligner, and count reads associated with mature miRNAs [30]. A strict cutoff of >100 counts was applied to filter miRNAs with potential physiologic relevance [31,32]. Log2-transformed data were used for differential expression analysis and functional prediction.

### 2.7. SARS-CoV-2 N Protein Serology

SARS-CoV-2 nucleocapsid (N) protein immunoglobulin G (IgG) was measured by ELISA using recombinant N protein. Briefly, microtiter plates (cat#9018, Corning Life Science, Corning, NY, USA) were coated with recombinant protein diluted to 2 µg/mL in 0.1 M carbonate buffer pH 9.4 overnight at 2–8 °C. Plates were washed and subsequently blocked with 3% milk powder blocking solution (cat# 232100, BD, Franklin Lakes, NJ, USA) for at least 1 h at room temperature (RT). After washing, serum samples were added to the plate in true duplicates diluted to a starting concentration of 1:100, followed by two-fold serial dilutions, and incubated for 2 h at RT. Following a wash, peroxidase-conjugated goat anti-human IgG (cat #109-035-008, Jackson Laboratories, Bar Harbor, ME, USA), diluted at 1:5000 in 1% blocking buffer, was added at a volume of 50 µL per well. After 1 h incubation at RT, plates were washed and developed using Sigma Fast o-phenylenediamine dichloride (OPD) (Sigma-Aldrich, St. Louis, MO, USA) for 15 min at RT. The reaction was stopped with 100 µL of 1 M sulfuric acid, and all plates were read at 490 nm. A 4-parameter logistic regression curve fit algorithm was used to calculate N-specific IgG concentrations using Softmax Pro 7.0.3 GxP software (Applied Biosystems, Waltham, MA, USA).

An in-house reference serum was developed by pooling convalescent serum from adult SARS-CoV-2-positive patients. This in-house reference serum was calibrated against the 2nd WHO International Standard for anti-SARS-CoV-2 Immunoglobin (NIBSC (21/340), Potters Bar, UK). The binding antibody unit (BAU) values assigned to the in-house reference serum for N protein was 325 BAU/mL. Serum samples collected prior to 2020 (n = 50) were used for the analysis of assay specificity. A value of 16 BAU/mL was selected as the threshold indicative of SARS-CoV-2 antibodies, based on the highest value of N protein IgG in samples from pre-COVID-19. The sensitivity of the assay to detect seropositivity at the selected threshold was 94.9% (37/39) for samples taken >14 days following the SARS-CoV-2-positive PCR test.

### 2.8. Bioinformatics and Data Analysis

Differential expression analysis, principal component analysis, and heatmap with unsupervised clustering were performed by in-house bioinformatic workflow (all scripts available via https://github.com/schuti/SVVATH-D, accessed on 12 December 2024) using R packages version 4.2.1. Functional/pathway enrichment analysis was performed by EnrichR (https://maayanlab.cloud/Enrichr/; last access on 2 January 2025). A predicted function with adjusted *p*-value < 0.05 was considered to be significant.

## 3. Results

A combination of stepwise centrifugation, ultrafiltration, and qEV-size exclusion chromatography (SC-UF-qEV) has been successfully employed in our previous studies to isolate EVs from cell culture media [28] and human plasma [33], resulting in an acceptable recovery yield and high specificity. This combined method has never been applied to human milk. It was noted that the human milk matrix is far more complex than cell culture media or plasma and contains numerous milk fat and casein micelles that could interfere with downstream EV analysis. We optimized the workflow for human milk by adding low-speed centrifugation to remove milk fat and 1% *w/v* sodium citrate, a well-known calcium chelator, to disrupt casein micelles as the early part of the SC-UF-qEV method (Figure 1a). Following the isolation procedure, we validated the presence of HMEVs in the isolates following the International Society for Extracellular Vesicles (ISEV) guidelines [34]. Nanoparticle tracking analysis (NTA) was performed to estimate the size distribution and concentration of the isolated HMEV particles. As shown in Figure 1b, the majority of particles in the HMEV isolate had diameters of <200 nm, consistent with the small EV subpopulation as defined by the ISEV [34]. Next, we performed TEM with negative staining to confirm the nano-scaled vesicle morphology (Figure 1c). Finally, Western immunoblotting was performed to detect three common EV markers: CD9, a surface tetraspanin; ALIX, a membrane-associated protein involved in EV biogenesis; and HSP70, a cytosolic protein commonly presented in EVs that is enriched in the HMEV isolate but not in human milk (HM), defatted HM (dHM), or microvesicles (MVs) (Figure 1d; full-length blots of the cropped images are provided in Appendix A).

We previously proposed that proteomics with unsupervised machine learning, i.e., principal component analysis (PCA), could be used as part of EV quality control [37]. To further differentiate the HMEVs from HM, defatted HM (dHM), and microvesicles (MVs), we performed a principal component analysis (PCA) based on 928 proteins detected in HMEVs, HM, dHM, and MVs. As seen in Figure 1d, the PCA results effectively separated the HMEVs from the other sample types. HM and dHM were more similar. MVs, or large EV subpopulations with a typical diameter of 200–1000 nm [34], were different from both HMEVs, HM, and dHM. The targeted analysis of selected proteins demonstrated the enrichment of three EV markers (Alix, Hsp70, and CD9), as well as lactadherin, butyrophilin, and xanthine dehydrogenase (XDH). These markers had been proposed previously [35,36] to be enriched in HMEVs but not in other milk fractions (Figure 1f), thus providing a potential explanation for the PCA result. These findings strongly support the use of this optimized workflow for further HMEV studies.

Mothers in the IMPRINT birth cohort were selected based on their reported history of COVID-19 infection during pregnancy from mid-2021 to early 2022 (SARS-CoV-2 Delta variant predominance). Their self-report was then confirmed by testing the third-trimester maternal serum sample for SARS-CoV-2 N protein IgG by ELISA. The combination of self-report and serology resulted in nine mothers with a history of SARS-CoV-2 infection during pregnancy (the prenatal SARS-CoV-2 group) and six healthy mothers (the control group) free of SARS-CoV-2 infection who had milk samples collected at 2 weeks of lactation (day 7–9 postpartum), which was used for HMEV isolation. The prenatal SARS-CoV-2 group had a significantly higher anti-SARS-CoV-2 N protein IgG (median [IQR]: 57.5 [38.7, 83.1] BAU) as compared to the control group (median [IQR]: 1.0 [0.6, 1.8] BAU) (Table 1). Besides the serological results of the anti-SARS-CoV-2 antibody, there were no statistical differences between groups with regard to demographic data and HMEV parameters, including particle numbers, protein, and RNA content (Table 1). None of the mothers had received the COVID-19 vaccine before milk collection. Therefore, analyzing HMEVs collected at 2 weeks of lactation provided an opportunity to test relatively long-term influences of post-acute SARS-CoV-2 infection on the mammary gland and lactation physiology.

The HMEVs isolated from nine prenatal-SARS-CoV-2 and six control mothers were analyzed by proteomics (Figure 2). A total of 1146 proteins were identified and quantified in HMEVs (full data in Appendix A). As part of the quality control, targeted analysis of the common EV markers (tetraspanins CD9, CD63, CD81, Alix, Hsp70) and HMEV-specific markers (lactadherin, butyrophilin, XDH), as well as angiotensin-converting enzyme 2 (ACE2; a known receptor of SARS-CoV2), was comparable between the prenatal-SARS-CoV-2 and control groups (Appendix A).

Following differential expression analysis, 52 significantly altered proteins (25 upregulated and 27 downregulated; *p*-value < 0.05) were analyzed by heatmap with unsupervised clustering (Figure 2a). To obtain the relevant HMEV proteins for functional enrichment analysis, a volcano plot with the thresholds of 2× fold change and *p*-value < 0.05 was applied (Figure 2b), resulting in 35 relevant proteins that were used for functional prediction. Functional/pathway enrichment analysis against three databases, i.e., GO biological process, KEGG, and reactome pathways (Figure 2c and Appendix A), revealed that the altered HMEV proteins may trigger functional changes in recipient cells following EV cargo transfer, specifically (i) metabolic reprogramming: alterations in GPI, PYGB, and AMY1A cytosolic enzymes involved in carbohydrate metabolism; (ii) mucosal epithelium development: upregulation of GRB2 and PIGR, which play roles in epidermal growth factor receptor signaling; and (iii) immunomodulation: upregulation of IGLL5, IGKC, IGLC2, IGHA1, and IGHA2 may modulate B-cell activation and antibacterial humoral immune response.

A total of 2588 unique mature miRNA transcripts with at least one count were identified across 15 individual HMEV samples. Of these, only 232 miRNAs passed a strict cutoff of >100 counts, representing high- to mid-abundance HMEV-miRNAs with potential physiologic relevance [31,32]. Interestingly, PCA and differential expression analysis showed no significantly different miRNAs between the prenatal SARS-CoV-2 and control groups (Figure 3a–c), suggesting a tightly regulated mechanism of miRNA packaging of mammary epithelial cells to preserve HMEV essential functions. Figure 3d shows the top 20 highest abundance miRNAs that determine HMEV functionalities (full expression data in Appendix A). For instance, we noted miR-148a-5p to be the most abundant HMEV-derived microRNA [5,38] and that it can influence various functions, including DNA methylation through DNA methyltransferase 1 [39], cell and tissue development [40], inflammation [41,42], and neuroprotection [43,44]. Furthermore, miR-146b-5p is known to regulate mammary alveolar cell development [45] and play crucial roles in innate immunity and inflammation [46,47].

## 4. Discussion

SARS-CoV-2 infection can lead to post-infection consequences that affect multiple organ systems. This study demonstrates that prenatal SARS-CoV-2 infection may affect the mammary gland; thereby, it alters the HMEVs released into breastmilk postpartum. These changes could modulate biochemical communication between prenatal SARS-CoV-2-infected mothers and their breastfed infants.

Previous studies have shown that HMEVs carry selective biomolecular cargo from the mammary gland that has the potential to modulate gene expression and cell signaling in infant tissues [5,6,7]. HMEVs represent a vital mechanism of mother-to-child communication, with current evidence suggesting that their primary role is vertical biochemical transmission [5,6,7]. The infant intestinal mucosa appears to be the primary target of HMEVs. In murine models, orally administered bovine milk EVs resist harsh gastrointestinal conditions and are subsequently detected in multiple distant organs [10,11], including embryos in utero [10]. Oral gavage of bovine milk EVs increases placental weight and promotes embryo growth and survival in mice [10]. Notably, these functions are linked to EV proteins and miRNA cargo. HMEVs comprise biologically active receptors, ligands, enzymes, and miRNAs, which can reprogram recipient cells [5,48]. Therefore, HMEVs could similarly elicit physiological responses in breastfed infants.

The current study shows that prenatal SARS-CoV-2 infection altered HMEV-derived proteins, but not miRNAs, at 2 weeks of lactation. Upregulation of proteins can regulate metabolism, mucosal tissue development, and B-cell immunomodulation (Figure 2). These can be interpreted as the enhanced mucosal site-specific functionalities of HMEVs following prenatal SARS-CoV-2 infection, thereby offering protection to breastfed infants against intestinal and respiratory diseases. HMEVs have been shown to exhibit broad antiviral effects in vitro (i.e., HIV, rotavirus, RSV, CMV, Zika) [49,50,51,52]. HMEVs may also attenuate viral-induced inflammation through miR-146b-5p, one of the most abundant miRNAs in HMEVs (Figure 3d). miR-146b-5p acts through the negative feedback inhibition of nuclear factor kappa B (NF-kB)-mediated inflammation; it mitigates signals from tumor necrosis factor receptor and Toll-like receptor/interleukin-1 receptor superfamilies [46,47], dampening excessive immune responses and suppressing severe inflammation. In this direction, HMEVs from prenatal SARS-CoV-2 mothers may boost mucosal site-specific functions to prevent severe respiratory viral infections, including SARS-CoV-2, in breastfed infants.

This study has limitations. First, this study had a relatively small sample size. Also, it is possible that the lack of observed differential abundance of HMEV-derived miRNAs could be related to heterogeneous sample groups. Given that miRNAs are highly specific and can be influenced by numerous factors, including biological variability, the heterogeneity of the sample population could affect the sensitivity of miRNA analysis in detecting subtle differences. Future studies using a larger cohort are required to confirm the findings. Second, this study analyzed HMEVs only at 2 weeks of lactation. It is known that HMEV molecules change across lactation [53,54,55]. A longitudinal study using human milk samples across the periods of lactation would address whether prenatal SARS-CoV-2 infection causes HMEV molecular changes temporarily or throughout lactation. Third, this study predicted that prenatal SARS-CoV-2 infection would modulate biochemical communication between mothers and breastfed infants through HMEVs, i.e., immunomodulation and altered mucosal tissue development. Moreover, clinical outcomes such as respiratory and gastrointestinal tract infections should be further evaluated by cohort studies focusing on breastfed infants. Fourth, this study could not analyze HMEV alterations during acute SARS-CoV-2 infection, as this was beyond the scope of this study. Rather, we focused on post-infection consequences of SARS-CoV-2 during pregnancy that may affect mammary glands, resulting in HMEV molecular alterations during postpartum without interference from the acute-phase inflammatory process. The effect of maternal COVID-19 vaccination and exposure to both SARS-CoV-2 infection and vaccination on HMEV molecular cargo also remains to be determined. Finally, direct evidence of HMEV functions at cellular levels is lacking, including the HMEV molecular-function relationship with regard to tissue development, immunomodulation, and viral inhibition, and should be pursued in future studies.

## 5. Conclusions

This study communicated, for the first time, that prenatal SARS-CoV-2 infection affected HMEV molecular cargo in a way that promotes mucosal site-specific functions of HMEVs. These findings require further validation in a larger sample size and follow-up across multiple stages of lactation to determine whether HMEV alterations after prenatal SARS-CoV-2 infection are temporary or persistent. Further investigation of the HMEV molecular-function relationship is warranted. Addressing these issues could inform public health strategies and breastfeeding practices to improve child health outcomes in the post-COVID era.

## Figures and Tables

**Figure 1 cells-14-00284-f001:**
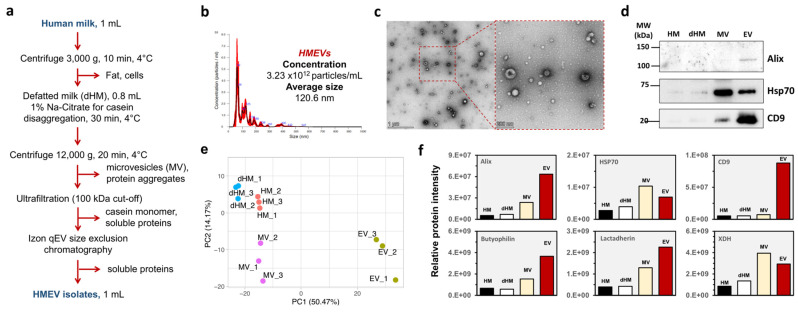
HMEV isolation and characterization. (**a**) The optimized HMEV isolation workflow. (**b**) Nanoparticle tracking analysis (NTA). (**c**) Transmission electron microscope (TEM). Scale bars, 1000 nm (left panel) and 200 nm (right panel), respectively. (**d**) Western immunoblotting against three common EV markers: Alix, Hsp70, and CD9. (**e**) Proteomics with PCA clearly separated HMEVs from HM, dHM, and MVs. (**f**) Relative protein intensities of three common EV markers and HMEV-specific markers [35,36], including butyrophilin, lactadherin, and XDH. dHM, defatted HM; Hsp70, heat shock protein 70; MV, microvesicle; PCA, principal component analysis; XDH, xanthine dehydrogenase.

**Figure 2 cells-14-00284-f002:**
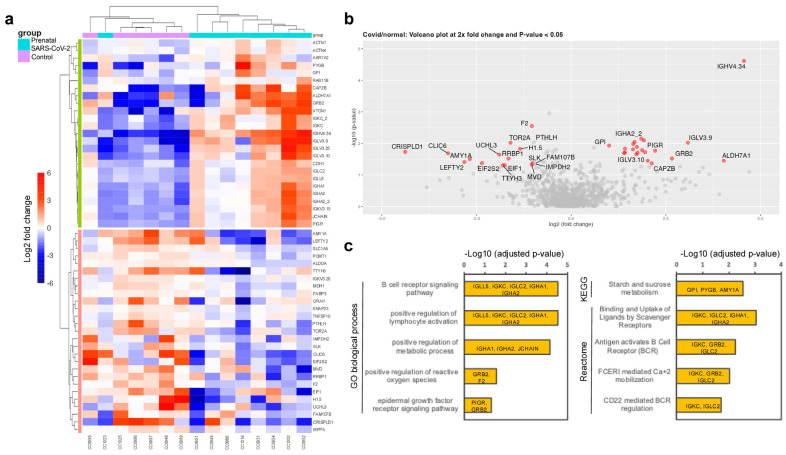
HMEV proteomic analysis revealed prenatal SARS-CoV-2 infection affected HMEV proteins at 2 weeks of lactation period. (**a**) Heatmap with unsupervised clustering of 52 significant proteins. Color scale represents protein fold-change over the median protein expression after log2 transformation. (**b**) Volcano plot showing 35 relevant proteins at the thresholds of 2× fold change and *p*-value < 0.05. (**c**) Functional enrichment analysis of 35 HMEV-relevant proteins against GO biological process, KEGG, and reactome. Adjusted *p*-value < 0.05 determined functional significance.

**Figure 3 cells-14-00284-f003:**
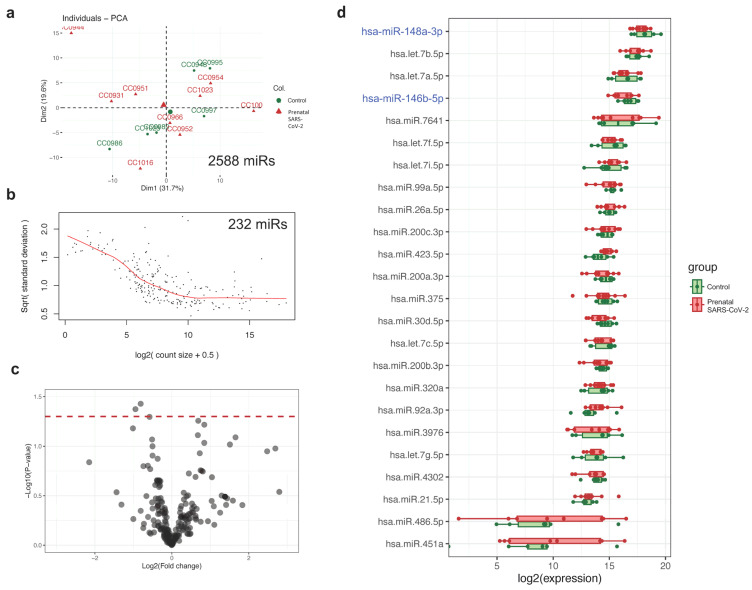
HMEV miRNA analysis in prenatal SARS-CoV-2 infection versus control groups. (**a**) PCA based on all 2588 identified miRNAs. A total of 232 miRNAs passed a cutoff of 100 counts. (**b**) Mean-variance trend plot for quality control. (**c**) Volcano plot shows no differentially expressed miRNAs between groups. (**d**) Boxplot showing the read counts of the 20 most abundant miRNAs in HMEVs.

**Table 1 cells-14-00284-t001:** Clinical variables and human milk characteristics.

	Prenatal SARS-CoV-2 (n = 9)	Controls (n = 6)	*p*-Value
Maternal variables			
Age (yr), mean ± SD	31.7 ± 5.2	31.5 ± 4.2	0.95
BMI (kg/m^2^), mean ± SD	27.8 ± 6.0	24.1 ± 4.2	0.22
Parity (n), median [IQR]	3 [2, 4]	2 [2, 3]	0.74
Time from COVID-19 test positive to delivery (week), median [min–max]	13 [9–16]	−	−
Serum anti-SARS-CoV-2 N protein IgG (BAU/mL), median [IQR]	57.5 [38.7, 83.1]	1.0 [0.6, 1.8]	<0.01
Infant variables			
Gestational age (week), mean ± SD	38.5 ± 0.7	38.6 ± 1.0	0.81
Birth weight (kg), mean ± SD	3.4 ± 0.3	3.3 ± 0.4	0.70
Human milk, 1 mL			
Protein concentration (after fat removal) (mg/mL), mean ± SD	16.9 ± 3.5	16.3 ± 2.4	0.70
EV particle concentration (10^11^ particles/mL), median [IQR]	5.3 [2.7, 6.1]	3.4 [2.0, 4.0]	0.34
Total EV protein (μg), median [IQR]	190 [148, 235]	159 [135, 211]	0.41
Total EV RNA (ng), median [IQR]	180 [105, 199]	175 [54, 473]	0.27

Abbreviations: BMI, body mass index; EV, extracellular vesicles; IQR, interquartile range.

## Data Availability

The original contributions presented in this study are included in the article/Appendix A. Further inquiries can be directed to the corresponding authors.

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
