# Peer review of "Prenatal SARS-CoV-2 Infection Alters Human Milk-Derived Extracellular Vesicles"

_cells, 2025, doi:10.3390/cells14040284_

Round 1
Reviewer 1 Report
Comments and Suggestions for Authors
The authors presented a study on the composition of milk exosomes in mothers with pre-natal SARSCov2 infection compared to a control group. There is limited research overall on how the composition of milk alters depending on pre-natal conditions hence in this regard, the study has significance. However, in my opinion, some issues need to be explained/addressed.
1. There is only Age, BMI, and Parity as clinical variables describing these mothers' pre-natal condition even though there are standard clinical variables used such as gestational age, infant weight, etc. The fact that a small number of samples were used and these samples are potentially heterogenous decreases even more the statistical power of the study. Could the authors comment on this fact?
2. Considering the above statement the authors appear certain about the following concluding remark: "The current study shows that prenatal SARS-CoV-2 infection altered HMEV-derived proteins, but not miRNAs, at 2-week of lactation". miRNAs are type-specific molecules, Could it be that due to heterogeneous sample groups, there isn't a differential abundance for miRs?
3. There is a discrepancy between the text and Figure 2a. Please clarify where you based the unsupervised clustering. Was it based on protein intensities or fold change?
4. What is the reasoning for performing miRNA target analysis about SARS cov2 infection when the miRs that the authors included do not show differential abundance?
Kind Regards
Comments on the Quality of English Language
The quality of English is good.
Reviewer 2 Report
Comments and Suggestions for Authors
The manuscript of Dr. Somchai Chutipongtanate, Dr. Ardythe Morrow, and co-authors describes the composition of extracellular vesicles of breast milk isolated from sars-cov-2 infected mothers against controls.
The authors are proposing that Evs from mothers after infection by Sars-2-CoV-2 are protective for the infant against viral infection. All techniques and methods for Evs characterization are correctly described.
(1) The detection of 35 proteins differentially expressed is founding the authors’ hypothesis.
(2) The detection of miRNAs by RNA sequencing was not significant between the 2 groups. Moreover, the reported 5 top miRNAs are not confirmed by q-PRC. The authors are reporting conflicting results with miR-146a-5p and 146b-5p (see comment (4)). Consequently, the in silico data generated from the 5 miRNAs and the discussion section are largely speculative. This approach fits better in a review than in a research paper.
The manuscript could be published by Cells, if drastic improvements were made.
Notably, the authors should reduce the speculation to a minimum. Their data are interesting and do not need erratic hypotheses to deserve publication.
Major modifications
(1) I suggest suppressing the word « postpartum » from the title giving : Prenatal SARS-CoV-2 infection alters human milk derived extracellular vesicles.
(2) Figure 2a, the name on x and y scales are unreadable on my copy. Please provide an illustration with a better resolution.
(3) On 232 miRNA transcripts, you did not find any difference between the 2 groups (line-314). You can propose that miRNA packaging into exosomes is highly regulated. But this is the end of the result section. The following data focusing an in silico analysis on the 5 top miRNAs are difficult to understand. You have to take into account my comments on (4) and (5) to re-write a better demonstration. It may be wise to simply reserve this analysis to, for instance, a future review.
(4) As shown Figure 3 a and b, the authors are mixing up, miR-146b-5p with miR-146a-5p. Apparently, from the discussion section, the data are about miR-146a-5p. However, if it were miR-146b-5p, the authors could also discuss a role of this miRNA in immunity as well as in mammary alveolar cell regulation. This is a blunder that changes the perspective of the manuscript. Thanks to carefully check the final choice between 146a or b. The duplicated name of nodes on figure 3b create a risk of mistake. Please use a single name, the blue one is perfect for higher size and readability.
(5) On Figure 3c, the upper scale for log10 (p-value) is difficult to read on my copy. I think that this panel c showing GO, Kegg, Reactome pathways on 5 miRNAs which are not discriminant for the 2 groups of breast milk under study, is counterproductive. Please consider improving justification or simply remove the information. The high amount of these 5 miRNAs against the other 227 miRNAs (56%) is a weak argument. Type-I interferon is an historical data in virus infection, so your point of view is not new.
Minor modifications
(1) Under Western blot : Leammli buffer, replace with laemmli.
(2) On Figure 3b, no need to duplicate the node’s name (146b and 146a, for instance).
Round 2
Reviewer 1 Report
Comments and Suggestions for Authors
Thank you for accepting my concerns. Congratulations for your study.
Reviewer 2 Report
Comments and Suggestions for Authors
The authors have correctly addressed my comments. The manuscript can now be published in Cells.